# Targeting Neuroinflammation in Osteoarthritis with Intra-Articular Adelmidrol

**DOI:** 10.3390/biom12101453

**Published:** 2022-10-11

**Authors:** Francesca Guida, Monica Rocco, Livio Luongo, Pietro Persiani, Maria Chiara Vulpiani, Sveva Maria Nusca, Sabatino Maione, Flaminia Coluzzi

**Affiliations:** 1Department of Experimental Medicine, University of Campania Luigi Vanvitelli, 80100 Naples, Italy; 2Department of Surgical and Medical Science and Translational Medicine, Sapienza University of Rome, 00189 Rome, Italy; 3Unit of Anesthesia, Intensive Care and Pain Medicine, Sant’Andrea University Hospital, 00189 Rome, Italy; 4I.R.C.C.S. Neuromed, 86077 Pozzilli, Italy; 5Department of Anatomical, Histological, Forensic Medicine and Orthopedic Science, Sapienza University of Rome, 00161 Rome, Italy; 6Physical Medicine and Rehabilitation Unit, Sant’Andrea University Hospital, 00189 Rome, Italy; 7Department of Medical and Surgical Sciences and Biotechnologies, Sapienza University of Rome, 04100 Latina, Italy

**Keywords:** palmitoylethanolamide, adelmidrol, hyaluronic acid, visco-induction, osteoarthritis, neuroinflammation, mast cells, joint degeneration and pain

## Abstract

Neuroinflammation is an emerging therapeutic target in chronic degenerative and autoimmune diseases, such as osteoarthritis (OA) and rheumatoid arthritis. Mast cells (MCs) play a key role in the homeostasis of joints and the activation of MCs induces the release of a huge number of mediators, which fuel the fire of neuroinflammation. Particularly, synovial MCs release substances which accelerate the degradation of the extra-cellular matrix causing morphological joint changes and cartilage damage and inducing the proliferation of synovial fibroblasts, angiogenesis, and the sprouting of sensory nerve fibers, which mediate chronic pain. Palmitoylethanolamide (PEA) is a well-known MCs modulator, but in osteoarthritic joints, its levels are significantly reduced. Adelmidrol, a synthetic derivate of azelaic acid belonging to the ALIAmides family, is a PEA enhancer. Preclinical and clinical investigations showed that the intra-articular administration of Adelmidrol significantly reduced MC infiltration, pro-inflammatory cytokine release, and cartilage degeneration. The combination of 1% high molecular weight hyaluronic acid and 2% Adelmidrol has been effectively used for knee osteoarthritis and, a significant improvement in analgesia and functionality has been recorded.

## 1. Introduction

Osteoarthritis (OA) is a leading cause of musculoskeletal chronic pain, affecting nearly 16% of the population, characterized by cartilage degeneration and stiffness. A mechanism-based approach to OA should include at least three targets: the peripheral mechanisms of inflammation, the central mechanisms of pain hypersensitivity, and the prevention of joint destruction [1]. In the last few years, mast cells (MCs) have been widely investigated as a possible target to be modulated for controlling peripheral and central nervous system inflammation [2,3].

Joints are a target of neuroinflammation in chronic degenerative and autoimmune diseases, such as OA and rheumatoid arthritis, and MC activation has been recognized as a prominent feature of the synovial tissue in patients with OA [4]. The aim of this review is to present preclinical and clinical evidence of a new intra-articular formulation containing Adelmidrol for targeting neuroinflammation in OA diseases.

## 2. Neuro-Immune Mechanisms Underlying OA

The pathogenesis of OA is correlated with different interrelated mechanisms including inflammation, neuroplasticity, cartilage disruption, and bone damage. Neuro-immune signaling may occur when innate immune cells produce algogenic factors that act on the pain pathway [5]. Indeed, during OA progression, the nociceptors innervating joints can be sensitized by locally generated mediators, such as inflammatory cytokines and chemokines, nerve growth factor (NGF) [6], and disease-associated molecular patterns (DAMPs) [7,8] (Figure 1).

The pathogenesis of OA is largely determined by the disrupted balance of pro- and anti-inflammatory mediators leading to a chronic low-grade inflammation which contributes to the associated pain condition.

The enhancement of proinflammatory cytokines causes the secretion of enzymes and other inflammatory factors involved in the pathogenesis of OA responsible for morphological joint changes, such as cartilage degeneration, osteophyte formation, and synovitis. One of the most important inflammatory mediators involved in this process is the interleukin (IL)-1β, a member of the IL-1 superfamily implicated in the pathogenesis of numerous neuroinflammatory diseases [9].

In OA, through the activation of the IL-1 receptor I (IL-1RI), IL-1β mediates cartilage destruction by inhibiting new formations of proteoglycans and degrading existing proteoglycans by stimulating degradative enzyme production. Widely distributed in different cells in the knee joint, IL-1RI has been shown to be upregulated in isolated human OA chondrocytes in vitro [10,11]. In OA synovium, a relative deficit in the production of natural antagonists of the IL-1 receptor (IL-1Ra) has been detected and it could possibly be related to the additional production of nitric oxide in OA tissues [12]. IL-1β activates several signaling pathways which overall contribute to the progression of OA. In particular, through mitogen-activated protein kinase (MAPK) signaling, IL-1β induces catabolic events including the cartilage extracellular matrix (ECM) degradation mediated by metalloproteinases (MMPs), such as MMP-1, MMP-3, and MMP-13, and aggrecanases such as ADAMTS-4 and ADAMTS-5. Additionally, the IL-1β-mediated NF-κB pathway activation leads to the synthesis and secretion of proinflammatory cytokines [IL-6 and tumor necrosis factor (TNF)-α], and chemokines [IL-8, monocyte chemoattractant protein-1 (MCP-1 or CCL2), CCL5 (RANTES), and macrophage inflammatory protein-1a (MIP-1a)] [13]. In OA, higher concentrations of IL-6 in synovial fluid have been correlated with the pain experience of OA patients [14] and in an experimental study, de Hooge et al. [15] showed that IL-6 gene knock out mice developed more advanced OA, compared to control animals. 

High concentrations of TNF-α have been observed in structures such as synovial fluid, synovial membrane, cartilage, and subchondral bone layer [16]. As with IL-1β, TNF-α, via its conjugate receptors (TNFR-1 and TNFR-2), promotes the activation of the JNK kinase and the transcription factor NF-κB [16], by leading to the degeneration of cartilage and morphological joint changes. 

Chemokines contribute to the inflammatory processes, stimulating the chemotaxis of inflammatory cells that, in turn, secrete proinflammatory cytokines, by representing thus the major challenge in treating and slowing the progression of osteoarthritis [11]. Increased levels of CCL2 have been detected in the synovial fluid of patients with knee OA [17]. Borzi et al. [18] demonstrated that CCL2 induces MMP-3 expression enhancement, contributing to proteoglycan loss and the degradation of cartilaginous tissue. In an experimental study, Miller et al., [19] showed that both CCL2 and CCR2 were upregulated in the innervating dorsal root ganglia (DRG) of the knee 8 weeks after the surgical model of OA while the absence of CCL2 or CCR2 suppresses selective inflammatory genes in joints 6 h post-de-stabilization of the medial meniscus (DMM) model, including arginase 1, prostaglandin synthase 2, nitric oxide synthase 2, IL-6, MMP-3, and the tissue inhibitor of metalloproteinase (TIMP)-1, by delaying pain development. CCL3 and CCL5 levels have been shown to be elevated in OA synovial fluid when compared with controls [17,20]. IL-1β-incubated human chondrocytes showed a significant upregulation of CCL3, CCL4, and CCL5 [21]. 

Chemokines from the CXC family also participate in the pathogenesis of OA. CXCL8 (IL-8) has been shown to be upregulated in the serum and in the synovial fluid of OA patients [22], and the study conducted by Guo et al. [23] reported that CXCL12 levels in the synovial fluid were closely related to the radiographic severity of OA. Interestingly, experimental evidence suggests a role for fractalkine (CX3CL1) and its receptor (CX3CR1) in the development of chronic pain in OA. Indeed, after DMM surgery, fractalkine release by DRG neurons is upregulated in the late phase of the model and this timing correlates with the development of microgliosis in the dorsal horn, where microglial cells express CX3CR1 [24]. The involvement of fractalkine and its receptor CX3CR1 in several pre-clinical models of chronic and neuropathic pain has been previously described [25,26]. In particular, this pathway has been highlighted as one of the most important cross-talk pathways involved in neuron-microglia communication [27].

As a member of the neurotrophin family, NGF is essential for the development of nociceptive primary neurons. NGF can bind the general neurotrophin receptor p75, as well its high-affinity cognate receptor, tropomyosin-related kinase (Trk)-A [28], which are expressed by joint cells, including chondrocytes, and increased in OA cartilage [29]. Triggered by a proinflammatory environment, the NGF synthesis is highly correlated with the degree of OA cartilage degradation in humans [29]. The NGF/TrkA signaling activation induces the release of a variety of inflammatory mediators such as serotonin, histamine, and NGF itself, which are known to cause sensitization of nociceptors. The injection of an anti-NGF monoclonal antibody shortly after model induction (day 2) was able to reverse the deficits in burrowing one day later compared to saline [30]. Moreover, the effects of a long-term NGF blockade in rat meniscal surgery models have been reported [31]. The authors showed that therapeutic treatment (day 14–day 21) reversed weight-bearing asymmetry and mechanical allodynia of the hind paw, while no short-term effects on histologic cartilage degeneration were observed. Similar findings were reported using Tanuzemab in a 28-day rat medial meniscal tear (MMT) model [32], where the authors described a significant increase in tibial cartilage degeneration, subchondral bone sclerosis, and tibial osteophytes induced by the treatment. 

Damage-associated molecular patterns (DAMPs) are a group of molecules released from ECM to the joint cavity during cartilage degradation. Together with so-called pathogen-associated molecular patterns (PAMPs), DAMPs signal to the immune system a “dangerous state” requiring a protective response to a pathological state. Indeed, joint trauma results in the production of DAMPs and intracellular alarmins that signal through pattern recognition receptors (PRRs), which include Toll-like receptors (TLRs) on synovial macrophages, fibroblast-like synoviocytes (FLS), or chondrocytes to induce the local production of inflammatory mediators of various factors, like catabolic factors [matrix metalloproteinase, cytokines, chemokines cathepsins (B, K, and L)], and the complement cascade [7,33].

High levels of NGF, released by MCs, influence sensory nerve activity and its interaction with the high-affinity TrkA receptor (NGF/TrkA), and amplify inflammatory processes contributing to the development of a nerve sprouting of sensory pain fibers which mediate chronic pain maintenance.

## 3. Role of Mast Cells in OA

The important role of MCs in allergic reactions is well-known. Moreover, the involvement of MCs in physiological bone turnover and bone disorders has been described in detail [34,35]. MCs usually reside quietly within tissues throughout the body, whereas when they become activated, they secrete granules containing histamine and other inflammatory substances. The hyperactivation of MCs is often accompanied by pain syndromes [36]. Indeed, pain-like behaviors have been found to be MC-associated in several experimental models of pain [37].

In healthy joints, synovial tissue macrophages (STM) or MCs represent nearly 3% of the cellular population of the synovia and act as sentinels, monitoring possible pathogens and mediating the immune response [38]. In chronically inflamed joints, the synovial membrane becomes hypertrophic and the MC density significantly increases, as a result of the recruitment of circulating progenitors, the maturation of local MC precursors, the differentiation of local mesenchymal stem cells (MSCs) into MCs, and the release of neurotrophins as NGF that stimulate further MC proliferation. Farinelli et al. have recently shown, in hip and knee tissue from OA patients undergoing arthroplasty, a significant increase in the MCs and the vessel number, which significantly correlated with the synovitis score and disease severity [39]. The synovial MC infiltration has been associated with a variety of inflammatory changes which may play a role in the pathophysiology of OA [40,41]. The increased vascular permeability mediated by the histamine and prostaglandin D2 released by MCs, the recruitment of other circulating polymorphonuclear cells (PMN), such as leukocytes, and macrophage activation induced by cytokines and chemokines, contribute to the early phases of inflammatory arthritis [42]. Additionally, MCs may participate in joint destruction by the induction of MMPs from fibroblast-like synoviocytes, chondrocytes, and other cell types in the arthritic cartilage [42,43]. MCs have been detected even in osteophytes, where they play a role in accentuating the inflammatory pathology of OA [44]. Similarly, in the rheumatoid-arthritis (RA) joint, activated MCs release pro-inflammatory and immunomodulatory mediators (TNF-A, IL-6, IL-8, VEGF, histamine, heparin, tryptase) that induce angiogenesis, promote cartilage erosions, synoviocyte proliferation, and pain sensitization. Synovial MC concentration has been shown to correlate with the different stages of RA, particularly with the maintenance or lack thereof of the remission phase, suggesting a possible role as a biomarker to predict the severity and progression of the disease [45]. Cytokines and growth factors released during inflammation are the main mediators of the cross-talk between the subchondral bone, cartilage, and synovia. In vitro studies reveal that MCs synthesize and release NGF in response to antigen/IgE stimulation and it is involved in MCs maturation and degranulation [46]. High levels of NGF, released by MCs, induce the sprouting of sensory pain fibers and contribute to peripheral and central sensitization, which are the cornerstone of chronic pain maintenance [47]. The injection of complete Freund’s adjuvant (CFA) induces inflammation of the joint and sprouting of CGRP+ nerve fibers in the synovium. This sprouting is significantly attenuated by anti-NGF therapy [48]. NGF binds TrkA receptors on inflammatory cells inducing the release of inflammatory mediators, such as serotonin, histamine, and even further NGF [49]. A more recent study suggested that NGF released from MCs may be implicated in osteoarthritis pain [50]. The authors have indicated the NGF responsible for MCs-PGD2-mediated increased nociceptive signaling, suggesting the TrkA blockade in MCs as a potential target for OA pain. Finally, MCs seem to play a role in the metabolism of hyaluronic acid (HA), which is the main component of the ECM and play a key role in joint lubrification during movement. MCs influence HA synthesis and degradation through two mechanisms of action. Firstly, MCs release tryptases, which may accelerate HA destruction. Secondly, granules of synovial MCs contain heparin, which represents the preliminary stage in the synthesis of HA. In vitro studies showed that MC chymase and tryptase accelerated the degradation of HA in the synovial fluid [51]. Moreover, MCs express the hyaluronic-acid-binding isoform of CD44 [52]. These observations could explain why the modulation of MCs could be a key factor for reducing intra-articular inflammation and ensuring the physiological metabolism of HA, filling the unmet needs of the currently available intra-articular therapies. 

## 4. Intra-Articular Adelmidrol: A New Therapeutic Option for OA

Intra-articular treatments in OA management are warranted as their effects are limited to damaged joints, without significant systemic adverse events. The most commonly used intra-articular treatments are high-molecular-weight HA, corticosteroids, leukocyte-poor platelet-rich plasma (LP-PRP), leukocyte-rich platelet-rich plasma (LR-PRP), bone marrow mesenchymal stem cells (BM-MSCs), and adipose mesenchymal stem cells (AD-MSCs) [53]. Intra-articular steroids are widely used for their anti-inflammatory effect. However, they have a mild to moderate effect on pain severity [54] and their chronic use has been associated with a number of clinical concerns, the most relevant being the alteration of gene expression and immunomodulatory effects, which are associated with additional joint damage [55]. Furth more, corticosteroids may inhibit the anabolic activity of chondrocytes, decrease collagen expression, and, when chronically used, be chondrotoxic [56].

Recent evidence showed that in symptomatic patients with knee OA, a combination of PRP and HA induces a significantly greater improvement of pain and function compared with HA injections only [57]. PRP infiltration in OA allows for the modulation of inflammation and the induction of cartilage regeneration, while HA is a pure visco-supplementation therapy, and is thus prone to degradation by proinflammatory cytokines, free radicals, and proteinases [58]. Although commonly used, this treatment presents extreme variability during the preparations, often influenced by donor-specific factors or by processing methods. As conventional treatments provide limited therapeutic benefits, novel strategies for managing OA are required.

Adelmidrol has been recently investigated as a possible innovative intra-articular therapy for OA. Adelmidrol is a synthetic derivate of azelaic acid which works as a palmitoylethanolamide (PEA) enhancer increasing endogenous PEA levels [59]. Recently, it has been demonstrated that Adelmidrol is able to significantly increase the endogenous PEA level in the duodenum and colon of healthy mice in a dose/time-dependent and PPAR-γ-independent manner [60]. As with PEA, Adelmidrol belongs to the family of Autacoid Local Injury Antagonist Amides (ALIAmides) and its amphiphilic and amphipathic characteristics make Adelmidrol particularly soluble and suitable for topical [61,62] and intra-articular administration. In healthy synovia, an elevated amount of PEA is present, while its levels are significantly reduced in the synovial fluid of patients with OA and rheumatoid arthritis [63]. In OA joints, the reduction of endogenous PEA levels is responsible for MC hyperactivation, which, by triggering a cascade of neuroinflammatory events, alters the physiological functionality of nerve endings, chondrocytes, and joint synvioblasts. Hypertrophic synvioblasts become unable to produce basic components of cartilage, leading to the progression of joint damage [42,64,65]. Therefore, the restoration of normal endogenous PEA levels, necessary to ensure MC normo-reactivity, becomes essential for the maintenance of normal joint function. These findings suggest that Adelmidrol, due to its properties, could represent a valid therapeutic approach for all diseases characterized by a significant reduction in endogenous PEA levels.

## 5. Adelmidrol in OA: Preclinical Evidence

The anti-inflammatory and immunomodulatory properties of both PEA and Adelmidrol have been shown in numerous experimental studies [66,67]. Particularly, Adelmidrol has been shown to reduce acute inflammation in the carrageenan (CAR) model, when systemically injected [68]. In the same study, the beneficial effect of Adelmidrol was also proved in collagen-induced arthritis (CIA), an animal model of rheumatoid arthritis. Adelmidrol was shown to reduce clinical signs of OA, such as periarticular erythema and paw edema. X-ray examinations have shown that Adelmidrol reduced bone erosions in the femoral growth plate, as well as in the tibiotarsal joints with a consequent reduction of pain and proinflammatory cytokines (TNF-α, IL-6, and IL-1β). Interestingly, the significant presence of MCs, with an increased chymase and tryptase expression in the joint tissues after CIA induction, was also reduced [68]. Preclinical observations of the combination of Adelmidrol and HA are available in a rat model injected with monosodium-iodoacetate (MIA) to induce OA. Di Paola et al. demonstrated that the combination of Adelmidrol with HA in a dose/time-dependent manner reduced the plasma levels of pro-inflammatory cytokines (TNF-a and IL-1b), MMPs, and NGF production in the osteoarthritic knee of the rats. Moreover, the combination significantly reduced MC infiltration and histological alterations induced by intra-articular MIA injection [69]. Among animal models developed for the examination of OA pathophysiology [70], MIA represents the most used model to study chronic pain associated with OA [71]. Along with joint damage, MIA injections induce mechanical sensitivity in the ipsilateral hind paw and weight-bearing deficits. Di Paola et al. observed that mechanical allodynia and motor functioning along with the degeneration of articular cartilage were reduced by the combination of HA and Adelmidrol [69].

## 6. Intra-Articular Adelmidrol: Clinical Evidence

A new intra-articular medical device containing Adelmidrol has been recently introduced onto the market (Hyadrol^®^, 2 mL pre-loaded syringe for intra-articular injection, Epitech Group SpA, Milano, Italy). In this product, 2% Adelmidrol is administered together with 1% high-molecular-weight (from 1.3 to 2.0 × 106 dalton) HA, which physiologically binds the CD44+ receptor [72]. Intra-articular Adelmidrol is supposed to be a valid therapeutic option for normalizing MC function and ensuring the visco-induction of HA, through a dual mechanism of action. Firstly, Adelmidrol, by increasing PEA levels in the OA joint, might switch MCs from the hyperactivated to the physiological status; thus, it may reduce the HA degradation, by inhibiting the release by MCs of lytic enzymes, such as tryptases, hyaluronidases, and β-hexosaminidases. Secondly, Adelmidrol, by normalizing the MCs’ release of heparin, which is a precursor of HA, induces HA production, leading to a visco-inductive effect in the OA joint (Figure 2).

A multi-center clinical trial has been published on the use of this combination of Adelmidrol and HA for intra-articular injection in patients suffering from Kellgren and Lawrence grade II-III osteoarthritis of the knee [73]. Eligible patients were ≥40-year-old subjects suffering from pain for more than 6 months, with an average pain score > 20 on the Western Ontario and McMaster Universities (WOMAC) Osteoarthritis Index pain sub-scale, and must have performed a washout of at least 2 weeks from other analgesics and/or non-steroidal anti-inflammatory drugs (NSAIDs). The principal exclusion criteria were: (a) the presence of concomitant severe systemic inflammatory joint diseases; (b) the use of corticosteroids in the previous 3 months; (c) the use of chondroprotective drugs, visco-supplementation injections, or arthroscopies within 6 months. The Italian WOMAC version was used in its VAS format: all 24 items were rated by the subject on a 10 cm VAS ranging from 0 (indicating no pain, stiffness, or functional limitation) to 10 (indicating extreme pain, stiffness, or functional limitation). Ranges of the WOMAC scores were: pain (0–50); stiffness (0–20), and functional limitation (0–170) [74]. The enrolled patients received 4 weekly intra-articular injections of 1% high-molecular-weight hyaluronic acid and 2% Adelmidrol in a 2 mL pre-loaded syringe. Patients were assessed using the WOMAC Osteoarthritis Index, the 12-Item Short Form Health Survey (SF-12) for the quality of life, and the Likert Patient Global Impression of Change (PGIC). In total, 102 patients (female n 60, 58.8%), with a mean age of 63.2 ± 0.6 years were recruited in this study, and follow-ups were conducted for 4 weeks after the last intra-articular injection. Only 7 patients withdrew prematurely from the trial: 3 for accidental trauma not related to the treatment, 3 due to knee pain and/or swelling after infiltration, and 1 was lost at the follow-up stage.

The 95 patients included in the final evaluation 4 weeks after treatment showed a significant improvement in all 3 elements of the WOMAC subscale (pain, stiffness, and functional limitation) (Table 1). Furthermore, a significant improvement in the quality of life was recorded in both the mental and physical components of the SF-12 (Figure 3).

Over 90% of the treated subjects reported a clinical improvement according to the PGIC subjective evaluation, among which 76.4% felt “very much” or “much” improved (Figure 4). Intra-articular injections of 1% high-molecular-weight hyaluronic acid and 2% Adelmidrol were well tolerated. No severe adverse events were observed. Seven patients (7.4%) reported local pain at the site of injection, accompanied by swelling in 2 patients (2.1%), regarded as related to the infiltrative procedure. Other adverse events, such as diarrhea, headache, abdominal cramps, and trauma were observed in 10 patients judged by the investigators as not related to the treatment [73].

Vulpiani et al. recently published the follow-up of the described clinical trial, limited to a group of patients enrolled in a single hospital (Sant’Andrea University Hospital, Rome, Italy), available for re-evaluation at 6 months, 1 year, and 2 years following the last intra-articular infiltration [75]. Forty-seven patients (77% of those enrolled in the first trial in a single center) were evaluated using both the WOMAC scale and the SF-12 questionnaire. The results obtained after 4 infiltrations, as measured with the WOMAC score, have been maintained for the whole follow-up period, including the effects on functional limitation. Similarly, no statistically significant modifications of both the physical and mental components of SF-12 have been observed in the 2-year follow-up period. These results suggest a long-term effect of Adelmidrol infiltration, which is not common for other intra-articular therapies.

## 7. Conclusions

OA is a complex degenerative disease, which requires a multimodal approach [1]. Systemic analgesia plays a role in targeting recurrent episodes of acute inflammation, which, by definition, lasts only a few days. NSAIDs may be useful for mitigating the dangerous effects of the destructive enzymes, cytokines, and prostaglandins released in the inflamed synovial membrane, after leukocyte infiltration [76,77]. Central analgesics, such as acetaminophen, opioids, and adjuvants, may be used for targeting maladaptive neuronal plasticity [78,79] and central mechanisms of pain [80]. Intra-articular therapies may support systemic analgesia by affecting the progression of joint destruction. In the real-world experience, however, there are still many unmet needs in the currently available intra-articular treatments for OA, in terms of analgesic efficacy, duration, tolerability, anti-inflammatory effect, and the modulation of cartilage degeneration.

Adelmidrol represents an innovative intra-articular treatment, which targets neuroinflammation and provides “visco-induction” to OA joints. It is time to “think outside the box”: we can prevent endogenous HA degradation, rather than merely provide HA through exogenous administration. Traditional exogenous HA is intra-articularly administered in order to provide “visco-supplementation” and to improve joint lubrification. However, when considering the inflammatory environment where the HA is injected, it is not surprising that its efficacy can be invalidated by the huge number of substances locally released, including MC-released tryptases and chymase, hyaluronidases, and β-hexosaminidases. Furthermore, MCs have played a key role in joint homeostasis, and their degranulation induces neuroinflammation.

MC-released pro-inflammatory cytokines fuel the fire of inflammation, promoting cartilage damage, angiogenesis, nerve sprouting, and pain sensitization.

PEA is a well-known substance that physiologically down-regulates MC activation and in OA joints, its endogenous levels are significantly reduced. Therefore, one of the targets of OA treatment, alongside the prevention of the progression of cartilage damage, should be to extinguish the fire of joint neuroinflammation.

Adelmidrol, by increasing and normalizing PEA levels in the joints, could be the first innovative molecule for “visco-induction”, through two mechanisms of action: (a) by reducing degranulation of MCs and therefore improving the efficacy of HA; (b) by normalizing MC function and restoring the physiological production of heparin, a HA precursor.

Preclinical and clinical data available on Adelmidrol administered together with HA support these hypotheses, by showing a significant reduction in cartilage degeneration, MC infiltration, and pro-inflammatory cytokine and chemokine plasma levels, and by improving analgesia and the functionality of OA patients.

Unfortunately, there are still many limitations to this emerging evidence. Firstly, further studies are needed to address the exact molecular mechanisms underlying the effects of Adelmidrol in the inflamed joint, despite the expected results demonstrated both in animal and clinical studies. Secondly, nowadays, clinical experience is still limited to the reported multicenter clinical trial by Vulpiani et al. [73] and the subsequent follow-up study [75], which confirmed the long-term results of this treatment. Therefore, further studies are warranted to confirm these interesting preliminary results. 

In conclusion, in the context of a multimodal approach to OA, where systemic analgesia plays a role in targeting recurrent acute inflammation and central neuroplasticity leading to chronic pain [1], intra-articular treatment with Adelmidrol can be considered a promising treatment for preventing the progression of joint degeneration and neuro-inflammatory processes.

## Figures and Tables

**Figure 1 biomolecules-12-01453-f001:**
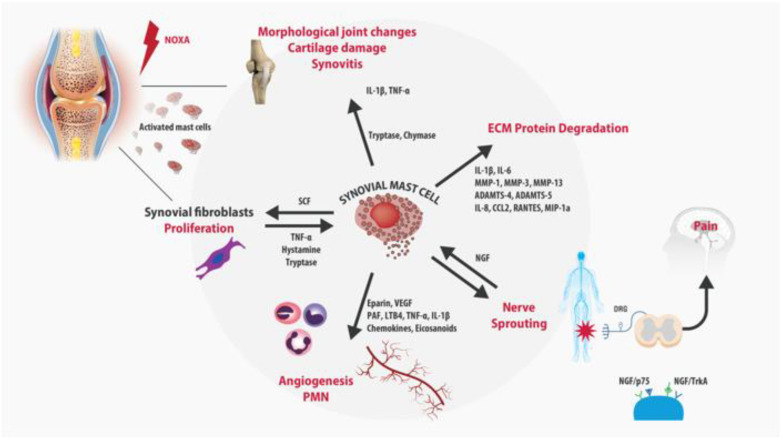
Schematic representation of the main contribution of MCs to degenerative joint diseases. Hyperactivated synovial MCs trigger the inflammatory process increasing vascular permeability and angiogenesis, recalling circulating PMN cells at the lesion site and activating synovial fibroblasts. Moreover, synovial MCs degranulation releases a massive amount of pro-inflammatory cytokines and chemokines, MMPs, and aggrecanases responsible for ECM protein degradation and, cartilage and bone remodeling.

**Figure 2 biomolecules-12-01453-f002:**
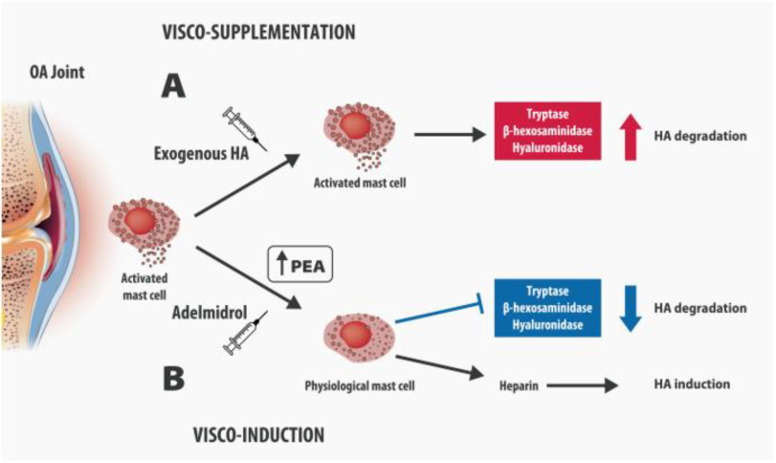
Visco-supplementation vs. Visco-induction. (**A**) Visco-supplementation: Exogenous HA is injected in the OA joints where activated MCs release tryptases, hyaluronidases, and β-hexosaminidases which in turn may accelerate HA degradation; (**B**) Visco-induction: Adelmidrol, by increasing endogenous PEA levels, downregulates MCs activation and shift them towards a physiological condition. Adelmidrol reduces the MCs degranulation, the release of lytic enzymes, and therefore the HA degradation, normalizing the physiological release of heparin, a HA precursor.

**Figure 3 biomolecules-12-01453-f003:**
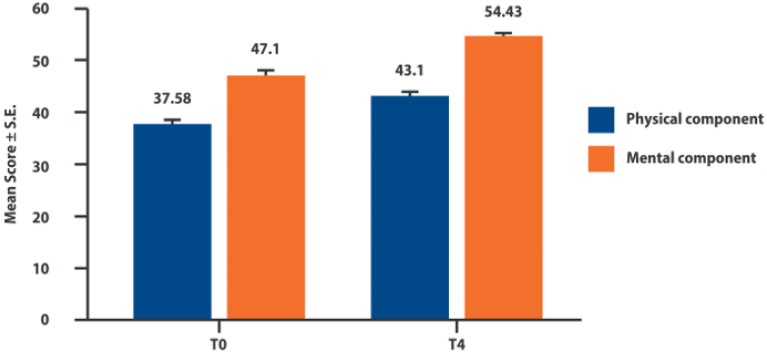
Changes in the SF12. Adapted from [73].

**Figure 4 biomolecules-12-01453-f004:**
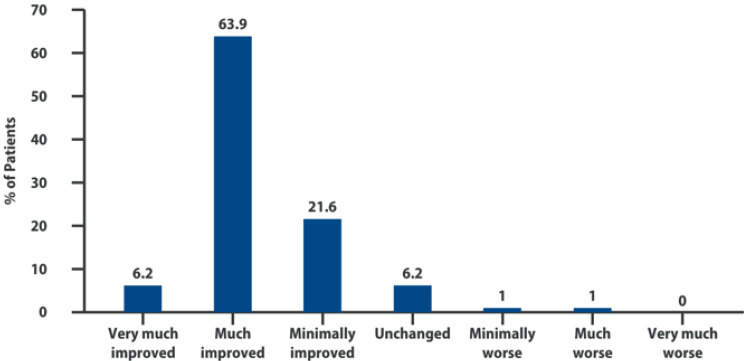
Patient Global Impression of Change. Adapted from [69].

**Table 1 biomolecules-12-01453-t001:** Changes in the WOMAC Osteoarthritis Index. Adapted from [73].

	T0	T1	T2	T3	T4	T5
	Basal (before 1st infiltration)	1st w	2nd ws	3rd ws	1st w after 4th infiltration	4th w after 4th infiltration
**N patients**	102	102	101	98	97	95
**Pain**	22.5 ± 0.76	16.8 ± 0.87	12.2 ± 0.81	9.4 ± 0.82	7.0 ± 0.76	5.2 ± 0.63
**Stiffness**	7.8 ± 0.49	6.3 ± 0.48	4.7 ± 0.43	3.6 ± 0.39	2.5 ± 0.33	1.9 ± 0.29
**Articular function**	67.3 ± 3.24	58.4 ± 3.10	46.3 ± 2.84	36.0 ± 2.83	28.1 ± 2.66	21.4 ± 2.13
**Total WOMAC score**	97.6 ± 4.11	81.5 ± 4.08	63.3 ± 3.78	49.1 ± 3.77	37.6 ± 3.53	28.5 ± 2.85

## Data Availability

Not applicable.

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
