# Peer review of "Targeting Neuroinflammation in Osteoarthritis with Intra-Articular Adelmidrol"

_biomolecules, 2022, doi:10.3390/biom12101453_

Round 1

Reviewer 1 Report

The review appears interesting, but my concern is that the author's perspectives on clinical aspects are largely based on the outcome of a clinical trial only in Italy. Though the study was a multicenter clinical trial in Italy to review Adelmidrol as a treatment for OA, it seems not significant enough to report it as an anti-neuroinflammatory drug. More literature or research data should be acknowledged or made available to qualify  Adelmidrol as an anti-neuroinflammatory drug for OA can consider its review for the drug.

Moreover, the introduction seems to be very precise. The authors must have discussed how Adelmidrol is superior in terms to NSAIDs and other drugs like Acetaminophen already used to treat pain for OA. Even except mast cells, other inflammatory cells like macrophages, fibroblasts, T cells, B cells, dendritic cells, and neutrophils do have a role in OA. The article has not discussed any role of  Adelmidrol on these cells to prevent neuroinflammation. 

Author Response

Q: The review appears interesting, but my concern is that the author's perspectives on clinical aspects are largely based on the outcome of a clinical trial only in Italy. Though the study was a multicenter clinical trial in Italy to review Adelmidrol as a treatment for OA, it seems not significant enough to report it as an anti-neuroinflammatory drug. More literature or research data should be acknowledged or made available to qualify Adelmidrol as an anti-neuroinflammatory drug for OA can consider its review for the drug.

A: We agree with the reviewer, as unfortunately nowadays there is only the multicenter clinical trial conducted in Italy. However, in the resubmission, we added the results, just published, of a follow-up clinical trial, which confirmed the long-term results obtained with Adelmidrol (see the new reference 75). Anyway, in the conclusions of the review, we modified the text underlying the limited clinical experience and suggesting the need for further investigations.

Q: Moreover, the introduction seems to be very precise. The authors must have discussed how Adelmidrol is superior in terms to NSAIDs and other drugs like Acetaminophen already used to treat pain for OA. Even except mast cells, other inflammatory cells like macrophages, fibroblasts, T cells, B cells, dendritic cells, and neutrophils do have a role in OA. The article has not discussed any role of Adelmidrol on these cells to prevent neuroinflammation. 

A: The aim of this review is to explain the rational by which Adelmidrol can be used as an alternative molecule to traditional intra-articular anti-inflammatory drugs, due to their well-known side effects in the joint. For this reason, authors did not consider the comparison of this device with systemic drugs, such as acetaminophen, which anyway play a role in the management of OA. Due to its route of administration, through intra-articular injection, adelmidrol can be considered an alternative to other commonly intra-articularly administrated drugs, despite direct comparison studies are still lacking. Authors insert a sentence in the conclusion chapter for explaining the role intra-articular treatments in the context of a multimodal approach to OA.

This review is focused on the role of mast cells, as this is the potential target of Adelmidrol, as a PEA enhancer. All the other cells involved in the pathogenesis of OA are partially affected by the modulation of mast cells, mediated by Adelmidrol, as clearly explained in the text.

Reviewer 2 Report

This is an interesting article merits publication. Some minor points are enlisted below

1. Line 86: should be TNFR1 and TNFR2. 

2. Line 120: [Iannone et al., 2002]. Could this reference numerically expressed in journal' style?

3. Line 219~239: The introduction about the linkage between Adelmidrol and OA should be extended. The impact of Adelmidrol on mast cells, the viscoinduction mechanism of  Adelmidrol, and the neuroinflammation-modulating mechanism should be introduced more to the readers.

4. Add more MDPI publications as reference.  

Author Response

Q: This is an interesting article merits publication. Some minor points are enlisted below

  1. Line 86: should be TNFR1 and TNFR2. 

A: Thanks for your suggestion. We modified the text accordingly.

  1. Line 120: [Iannone et al., 2002]. Could this reference numerically expressed in journal' style?

A: We apologize for the mistake. The reference has been corrected.

  1. Line 219~239: The introduction about the linkage between Adelmidrol and OA should be extended. The impact of Adelmidrol on mast cells, the viscoinduction mechanism of  Adelmidrol, and the neuroinflammation-modulating mechanism should be introduced more to the readers.

A: This part of the text has been expanded for better clarifying the relationship between Adelmidrol and mast cell, and the neuroinflammation-modulating mechanism (chapter 3). Dual mechanism of action of Adelmidrol has been clarified in depth in chapter 5.

  1. Add more MDPI publications as reference.  

A: These references, from MDPI journals, have been added:

Int. J. Mol. Sci. 202223(1), 541; https://doi.org/10.3390/ijms23010541

Int. J. Mol. Sci. 2019, 28, 2040, doi: 10.3390/ijms20082040.

Best Pract Res Clin Rheumatol. 2010 ,24, 301-12. doi: 10.1016/j.berh.2009.12.007.

Curr Rheumatol Rep. 2015, 17, 50. doi: 10.1007/s11926-015-0524-1.